# Indoor Environmental Quality Evaluation Strategy as an Upgrade (Renovation) Measure in a Historic Building Located in the Mediterranean Zone (Athens, Greece)

**Chrysanthi Efthymiou** [1,*][ID]**, Nikolaos Barmparesos** [1][ID]**, Panagiotis Tasios** [1]**, Vasileios Ntouros** [1][ID]**, Vasileios Zoulis** [1]**, Theoni Karlessi** [1]**, José Manuel Salmerón Lissén** [2][ID] **and Margarita Niki Assimakopoulos** [1][ID]

[1] Department of Applied Physics, Faculty of Physics, University of Athens, Building Physics 5, University Campus, 157 84 Athens, Greece; nikobar@phys.uoa.gr (N.B.); panagiotistasios@gmail.com (P.T.); vntouros@phys.uoa.gr (V.N.); sph1700037@uoa.gr (V.Z.); karlessith@phys.uoa.gr (T.K.); masim@phys.uoa.gr (M.N.A.)

[2] Grupo de Termotecnia, Escuela Técnica Superior de Ingeniería, Universidad de Sevilla, Camino de Los Descubrimientos S/n, 41092 Sevilla, Spain; jms@us.es

\* Correspondence: c-efthymiou@phys.uoa.gr; Tel.: +30-2107276731

**Abstract:** The assessment of indoor environmental quality in historic buildings converted to museums is a significant tool in deep energy renovation processes, as it provides insights for the microclimatic conditions in the interiors of the building where vast numbers of visitors walk every year and where artifacts that are vulnerable to pollution are exhibited. In this work, aiming to contribute to the development of an energy retrofitting protocol applied in the Mediterranean region (HAPPEN MedZeb protocol) for museums hosted in historic buildings by providing useful data, an experimental campaign to evaluate the indoor environmental quality of a museum housed in a historic building located in Athens took place from February 2019 to April 2021 and was divided into two periods. The findings revealed high concentrations of volatile organic compounds as well as poor thermal comfort levels since the sensors recorded low acceptable percentages of T values within the limits from 7 to 33% for the entire experimental period. Based on the findings, recommendations for retrofitting interventions are made.

**Keywords:** indoor environmental quality; historic buildings; museums; retrofit protocol; Mediterranean zone; HAPPEN project

## 1. Introduction

Indoor Environmental Quality (IEQ) in historic buildings such as museums is an important parameter to be considered during the design phase of a retrofit intervention. Microclimatic conditions in the interior spaces of a building and air pollution, either in the form of gaseous pollutants or in the form of particulate matter, impact the conservation of various artifacts as well as the physical and mental health of visitors and employees [1]. Due to the delicate nature of the exhibits, the indoor environment of a museum should meet specific hygrothermal and air quality requirements [2]. However, factors such as the diversity in collections and their respective conservation requirements, the fluctuating external thermal loads as a result of changes in weather patterns or daylight, and the natural deterioration of the building in cases of historic heritage buildings impede the maintenance of the stable optimum indoor microclimatic conditions that are required for the optimal preservation of the in-house collections [3–5]. As a result, during the decision-making phase of a retrofitting strategy to be implemented in a historic building, the designer should take the vulnerable nature of the exhibits into strong consideration.

Indoor environmental quality in museums plays a vital role in the preservation and conservation of collections [6], and it is also a key factor for the well-being of the museum's employees and visitors [7]. Hygrothermal variations, high concentrations of volatile

organic compounds, particulate matter, dust, light sensitive pollutants, and various gaseous pollutants (i.e., $O_3$, $NO_2$, $SO_2$) have adverse effects on human health and can degrade the quality of the exhibits by causing damage. Respiratory problems, nausea, and headaches are among the most common reported health problems that humans experience due to poor air quality [8,9], whereas alterations in relative humidity and temperature as well as the effects of fine particles on exhibits are linked to various threats, with corrosion and discoloration being the most common damages to exhibits [10–12]. According to Ilies et al., it is recommended that artifacts be exhibited in rooms where the temperature ranges from 14 °C to 24 °C and where the relative humidity varies from 20% to 60% depending on the nature of the exhibits, and the concentration of carbon dioxide should not exceed 1000 ppm, which is the limiting level for human health [1,13]. In ASHRAE's 2015 Handbook, it is noted that within museums, art galleries, libraries and archives, the temperature is suggested to be set between 15–25 °C and for the relative humidity to be at 50%, while the relevant short fluctuations should not exceed ±15 °C and ±5%, respectively [14]. In the same reference, the concentration of specific volatile organic compounds within museums should be less than 100 ppb with regard to general collections, whereas the limiting values for key gaseous pollutants sensitive materials are 0.09–4.89 $\mu g\ m^{-3}$ for $NO_2$, 0.1–1.05 $\mu g\ m^{-3}$ for $SO_2$, and less than 0.1 $\mu g\ m^{-3}$ for $O_3$.

In particular, sulfur dioxide ($SO_2$) and nitrogen dioxide ($NO_2$), gaseous pollutants that can be converted to sulphuric and nitric and nitrous acids, respectively, can cause damage to marble, limestone, metal, paper, and leather and can lead to color changes [15,16]. Discoloration can also be caused when exhibits are exposed to prolonged periods to ozone ($O_3$), while photographic material and paper is also affected by $O_3$ [17,18]. Volatile organic compounds, which are emitted from chemicals required for the preservation of artifacts as well as protection from indoor building materials, not only harm human health but can have negative effects on exhibits in high concentrations [19,20]. Exposure to particulate matter and dust, apart from worsening respiratory and long-term cardiovascular problems in humans, also facilitate water condensation on object surfaces, which subsequently makes them more susceptible to damage [21,22] In addition, illuminesce should not exceed specific levels, as it poses a threat to exhibits such as embrittlement, yellowing, and overall weakening [23]. Therefore, stable optimum microclimatic conditions and improved indoor air quality should go hand in hand with energy efficiency when planning a renovation strategy in museums.

Nevertheless, most historic buildings were not designed initially to function as museums, and their original technical specifications do not always meet the rigorous requirements for artifact conservation. In addition, historic buildings are usually protected by country specific laws on cultural heritage; thus, interventions that jeopardize their architectural quality are not always allowed [24]. Thus, in order to control heat transfer and moisture fluctuations and to achieve adequate levels of thermal comfort and improved indoor air quality in an energy efficient manner, interventions should not be invasive, whereas their heating, ventilation and air conditioning (HVAC) system as well as any intervention on the building's envelope should be carefully designed [25,26].Ongoing climate change is not only a threat to the integrity of buildings, affecting their degradation rates due to the severity of weather phenomena, but also calls for retrofits in order for buildings to sustain optimum thermal comfort conditions [27]. Especially in regions such as the Mediterranean zone, where heat waves are expected to become more intense and last longer [28], museums should provide shelter not only to employees and their thousands of visitors, but to their artifacts as well.

Retrofitting approaches and upgrade measures in historic buildings and museums have been implemented with success during the last two decades within the Mediterranean and in other regions [29]. Pisello et al. presented a replicable method for renovating a historic building in Italy by combining both active and passive energy efficiency techniques, which led to average energy savings greater than 64% for heating and cooling [30], while in another work, an integrated approach for the restoration of heritage buildings by

means of combined passive and active retrofit strategies was proposed [31]. Interestingly, Zannis et al. conducted a pre- and a post-intervention measurement campaign on eight museums across Europe, in which various measures to improve the microclimatic conditions and the buildings' overall energy efficiency were adopted. As shown in the results, the internal thermal comfort conditions were improved while also respecting the specific requirements of the exhibits, and on the other hand, adequate amounts of energy were conserved, with savings ranging from 39% to 77% [32].

This paper aims to contribute to the previously discussed framework by undertaking an indoor evaluation strategy as an upgrade measure in a historic building located in the Mediterranean zone (Athens, Greece) to not only enhance the already existing literature concerning IEQ and thermal comfort regimes but also to provide useful data for the development of a state-of-the-art energy retrofitting protocol for museums hosted in historic buildings.

## 2. Methodology

### 2.1. Reasons for the Choice of the Pilot Building

The examined museum is an outstanding example of a cultural building and offers the opportunity to widen HAPPEN Project (H2020-EE-2016-2017) outreach to thematic building sectors. HAPPEN (Holistic APproach and Platform for the deep renovation of the med residential built Environment) Project aims to enhance the market of deep energy retrofitting, especially in Mediterranean countries, by using IAQ and thermal comfort and energy data in order to develop an optimum retrofit approach that has been tailored for the Mediterranean environments [33].

The MedZEB Protocol is a guarantee scheme that sets the quality conditions for the good execution of the retrofit process along the whole value chain. The protocol includes tools such as the Building Renovation Roadmap, which is based on the HAPPEN step-by-step approach—in which the building is considered as a whole to avoid lock-in effects—and the Business Plan, which based on the VEL financial solution, which allows managing both short and long-term renovation plans according to a pre-given and coherent design framework.

The typology met in the investigated building is common in historic and heritage buildings in the Mediterranean, providing a representative example for testing, validating, and disseminating the project's concepts and outputs. It offers high replication potential, as it is a representative example of the historic building type. This is related to all of the construction and renovation issues that arise in the typology of a certain building, including structural issues and restrictions to renovation aspects (special licenses due to preservation issues) that arise from their historic character. The challenge related to this aspect is to implement deep renovation that maintains the special characteristics of a specific building at the same time.

### 2.2. Building Description

The Athens University Museum (Figure 1), which is housed in the "Cleanthes/Schaubert Residence" or "Old University", is located in Plaka, a historical neighborhood in the center of Athens. It is a 3-storey cultural building with a total surface of 830 m$^2$. Its permanent exhibition includes memorabilia from the start of its operation (manuscripts, scientific instruments, portrait paintings, etc.) and also functions as an event space, especially during summer periods. In 1836, it opened its doors as the first university in Greece after its independence and operated as such until 1841.

Today, the building operates as a museum and cultural center and receives a large number of visitors daily from 9:00 to 16:00. The museum has 10 permanent personnel. Currently, there is no central building management system (BMS), and most of the systems operate manually. Moreover, the building in naturally ventilated with open doors and windows.

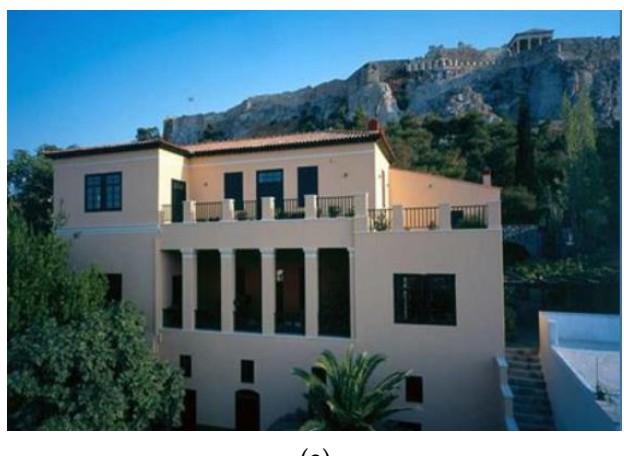
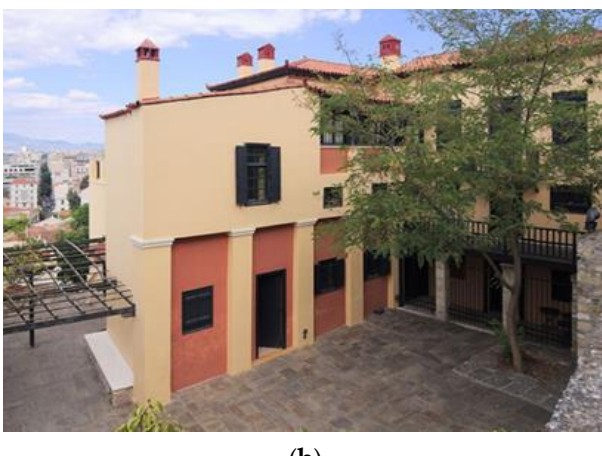

| (**a**) | (**b**) |

**Figure 1.** (**a**) Façade and (**b**) backyard of the Athens University Museum, Cleanthis-Schaubert Residence.

The main objective of a deep retrofitting on this building is to reduce energy consumption as well as to control the levels of temperature, humidity, indoor air quality (IAQ), and illuminance. Thus, the economic benefit will be significant. In addition, improved indoor air and thermal comfort levels will create a more pleasant microenvironment not only for visitors but also for the permanent staff, while the selection of a state-of-the-art lighting system will benefit the exhibits of the museum.

In more detail, the building consists of the ground floor with an event hall (Auditorium) and one warehouse, a 1st floor with five exhibition rooms, an intermediate floor with one office space, and a 2nd floor with two exhibition rooms and two offices. The main part of the building (walls, foundations) is made of stone with masonry mortar (U-Value = 2.89 W/m²K), with the total absence of insulation. There are underground retaining walls of reinforced concrete. The floors of the building are wooden. The balconies are wooden with reinforced iron bars. The roof is constructed from wood and is covered with tiles (U-Value = 3.05 W/m²K). Window frames are wooden with double glazing (U-Value = 3.04 W/m²K). Doors are also wooden. The heating (EER= 2) and cooling (EER = 3) system is based on electrical energy with the use of HVAC spilt units. As for the current state of the before-mentioned HVAC system, which can be summarized as old enough (installed 15 years ago), uses Freon R22, which is no longer on the market, several of them are out of order, there is no humidity control, and the system is not capable of keeping the temperature consistent. The electric energy consumption of the building according to the Public Power Corporation S.A-Hellas (ΔEH) reaches 76,240 kWh during the day and 19,520 kWh during the night.

### 2.3. Climate Aspects

The Greek National Regulation on the Energy Performance of Buildings—KENAK [34], places Athens in climatic Zone B; however, according to the Climate Severity Index (CSI) methodology implemented in the framework of the HAPPEN project [35], Athens is included in climatic zone W1S3 (winter zone 1 and summer zone 3), which is characterized by low CSI values for the winter (0.08) and high values for the summer (1.87). This means that summer is more "severe" in Athens, with elevated temperatures and high amounts of solar radiation, resulting in increased cooling needs compared to the heating demands for buildings, especially in the city center.

### 2.4. Monitoring Set-Up

The selection of all of the experimental points was based on specific criteria such as the type of usage for each room (exhibition rooms, offices, auditorium, etc.), their occupancy levels (staff only or also for visitors), and their position within the building (ground, first, second floor and outdoors). After the selection of the experimental positions,

10 environmental sensors (with data loggers) were installed in each separate room of the museum. More precisely, according to Figure 2:

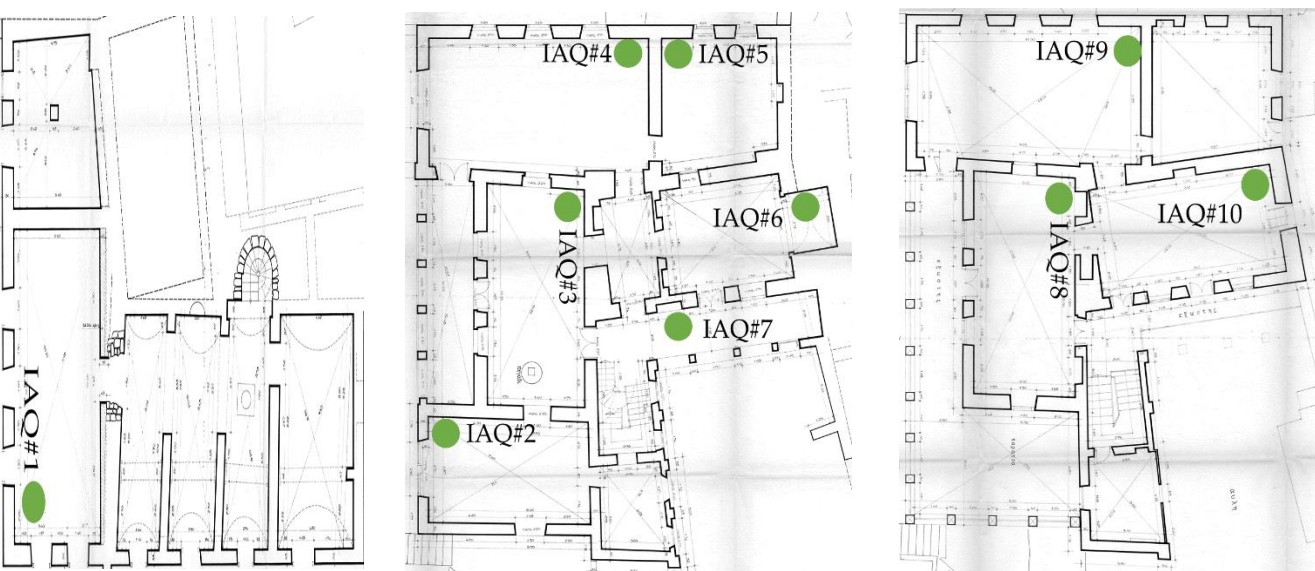

**Figure 2.** Selected experimental points within the museum. From left to right: ground floor, 1st floor, 2nd floor.

One sensor (IAQ#1) was placed in the Auditorium Hall of the ground floor.

Five sensors (IAQ#2–6) were situated within the exhibition rooms on the first floor (Entrance, Law, Medical, Presentation room, Dentistry).

One sensor (IAQ#7) was installed outdoors so that the influence of the ambient environment to the indoor examined parameters could be assessed.

Three sensors, two in the Philosophy and Applied Sciences exhibition rooms (IAQ#8 and #9) and one in an office (IAQ#10), were deployed on the second floor.

Thus, the influence of different visiting patterns and positioning within the building on the results could be investigated, differences between the public accessible halls and offices could be investigated as well. It should be noted that the experimental campaign was divided into two monitoring periods, 1 February 2019–26 August 2020 and 1 January–22 April 2021, in order to investigate the annual thermal behavior and IAQ levels of the investigated building.

### 2.5. Instrumentation

The equipment used in this experiment consisted of portable Tongdy sensors (Figure 3) that simultaneously recorded temperature (T), relative humidity (RH) as well as concentrations of carbon dioxide ($CO_2$) and total volatile organic compounds (TVOC). The $CO_2$ sensor ranged from 0 to 2000 ppm, with an accuracy of $\pm 40$ ppm at 25 °C, while the TVOC sensor ranged from 1 to 30 ppm, with an accuracy of 1 ppm. In addition, the measuring ranges for T and RH were 0 to 50 °C and 0 to 95% (non-condensing). All parameters were recorded on a 24 h basis at 15 min intervals and at least 1 m from the ground. Quality assurance for the equipment was performed on several occasions during the experiment, and all of the instruments were calibrated according to the manufacturers' standards.

It should also be mentioned that statistical differences between the groups of variables were tested for significance with the use of the Kruskal–Wallis-H test (for non-parametric data). The analysis showed that all *p*-values were <0.05, allowing us to reject the null hypothesis of no significant difference between the ranks of the grouping variables.

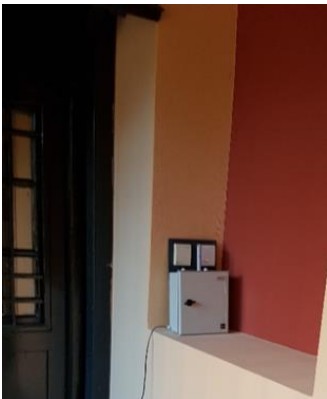

**Figure 3.** Air quality sensor situated within the building.

## 3. Results and Discussion

### 3.1. Thermal Conditions

Indoor and outdoor levels of T and RH were recorded for the entire experimental period in order to assess the thermal comfort conditions within the museum during warm and cold months. Thermal conditions within each room tended to fluctuate depending on the floor, the use of HVAC systems, and the time of year. Table 1 depicts the mean monthly results of the T and RH levels monitored in and out of the building and were divided per floor.

From this Table, one may notice that the highest mean monthly value of T was found on the second floor, reaching 31.95 °C in August 2020 (with ambient T of 30.96 °C), while the maximum T within the building (34.6 °C) was recorded on the same floor during the same month, with maximum outdoor T reaching 39.29 °C. The lowest mean value of T was recorded on the ground floor during January 2020 (14.67 °C), with an outdoor T of 10.37 °C, while amid the same month, the minimum value (9.56 °C) was found on the first floor (specifically in the Entrance), with the lowest ambient value reaching 6.67 °C. It is clear that during the hot summer months, the rooms on the second floor are the most vulnerable to elevated T levels, as they are directly exposed to large amounts of solar radiation due to the Earth's angle during this season. The building absorbs thermal energy from the sun and gradually yields it within as time passes. Thus, the preservation of the exhibits in the Philosophy and Applied Sciences rooms should be taken under careful consideration, especially during the warmer days of the year.

On the contrary, the lowest average T levels were monitored on the ground floor, during the colder months, which was mainly because no HVAC system operates in the Auditorium if it is occupied by visitors. As expected, the rooms on the first floor retain intermediate levels of T, which is mainly due to the operation of air conditioners as well as to the extra thermal insulation from the ground and the second floor. Nevertheless, as mentioned above, the lowest T value was recorded in the Entrance, which was mainly due the frequent opening of the door.

Referring to RH, the highest monthly average value was recorded on the third floor (64.44%) during November 2019, a month with frequent episodes of precipitation in the city of Athens, when the mean external RH was 69.83%. However, this is only an exception, as the regular trend of RH is to be elevated mainly on the ground floor for almost all of the experimental months. The non-frequent use of HVAC in this space does not favor dehumidification processes, and thus, elevated levels of RH appear.

Figures 4 and 5 summarize the results of internal and external T and RH for all of the investigated rooms during the entire experimental period. In both Figures, the coloured dotted lines represent the proposed limit ranges needed to achieve the desired thermal comfort conditions for the inhabitants, which were first reported in the research of Pegas et al. [36] and the National Building Regulations [37]. It should be noted that the proposed

T limits refer to the cold (from October to April, 20–23 °C) and warm periods (from May to September, 23–26 °C) of the year, while the limit range of RH (30–60%) is annual.

**Table 1.** Average monthly values of temperature (°C) and relative humidity (%) in and out of the building during the entire experimental period.

| Month | Parameter | Ground Floor | 1st Floor | 2nd Floor | Outdoor |
|---|---|---|---|---|---|
| | | | 2019 | | |
| February | T (°C) | 15.42 | 16.47 | 17.57 | 12.07 |
| | RH (%) | 52.13 | 50.52 | 47.30 | 63.13 |
| March | T (°C) | 16.75 | 17.12 | 18.39 | 15.97 |
| | RH (%) | 50.66 | 50.55 | 45.68 | 52.42 |
| April | T (°C) | 19.82 | 18.92 | 20.11 | 17.79 |
| | RH (%) | 50.46 | 51.42 | 48.70 | 56.10 |
| May | T (°C) | 21.50 | 21.44 | 23.92 | 22.15 |
| | RH (%) | 53.83 | 51.84 | 42.98 | 51.49 |
| June | T (°C) | 25.85 | 25.72 | 29.15 | 28.43 |
| | RH (%) | 52.49 | 51.22 | 41.71 | 44.68 |
| July | T (°C) | 26.11 | 26.15 | 27.85 | 30.15 |
| | RH (%) | 47.84 | 46.10 | 39.94 | 39.78 |
| August | T (°C) | 27.11 | 27.11 | 29.61 | 31.19 |
| | RH (%) | 40.30 | 43.94 | 36.96 | 35.53 |
| September | T (°C) | 25.40 | 26.63 | 28.12 | 27.43 |
| | RH (%) | 41.65 | 44.89 | 37.91 | 42.73 |
| October | T (°C) | 23.04 | 24.50 | 25.73 | 23.49 |
| | RH (%) | 53.36 | 53.03 | 45.47 | 55.29 |
| November | T (°C) | 20.72 | 21.62 | 20.38 | 19.18 |
| | RH (%) | 63.97 | 61.29 | 64.44 | 69.83 |
| December | T (°C) | 17.63 | 18.96 | 19.85 | 13.41 |
| | RH (%) | 55.57 | 53.21 | 51.30 | 54.90 |
| | | | 2020 | | |
| January | T (°C) | 14.67 | 16.48 | 18.29 | 10.37 |
| | RH (%) | 49.80 | 47.57 | 43.02 | 59.08 |
| February | T (°C) | 17.06 | 17.23 | 18.70 | 11.30 |
| | RH (%) | 48.50 | 49.93 | 45.20 | 54.90 |
| March | T (°C) | 16.33 | 17.36 | 18.74 | 16.44 |
| | RH (%) | 55.10 | 52.01 | 46.72 | 50.73 |
| April | T (°C) | 20.84 | 22.53 | 25.87 | 23.96 |
| | RH (%) | 53.84 | 50.38 | 41.24 | 45.99 |
| May | T (°C) | 23.22 | 24.73 | 27.77 | 26.88 |
| | RH (%) | 55.94 | 51.84 | 42.98 | 46.43 |
| June | T (°C) | 26.61 | 28.26 | 31.19 | 31.02 |
| | RH (%) | 49.03 | 45.28 | 37.00 | 38.00 |
| July | T (°C) | 27.63 | 29.30 | 31.91 | 30.93 |
| | RH (%) | 48.38 | 44.30 | 37.06 | 40.23 |
| August | T (°C) | 27.63 | 29.30 | 31.95 | 30.96 |
| | RH (%) | 48.38 | 44.30 | 37.06 | 40.23 |
| | | | 2021 | | |
| January | T (°C) | 15.28 | 16.46 | 16.81 | 13.28 |
| | RH (%) | 57.89 | 56.13 | 54.87 | 65.08 |
| February | T (°C) | 20.25 | 17.21 | 18.24 | 13.61 |
| | RH (%) | 44.99 | 52.80 | 49.61 | 63.22 |
| March | T (°C) | 16.66 | 17.52 | 17.76 | 15.04 |
| | RH (%) | 46.29 | 45.25 | 45.12 | 52.12 |
| April | T (°C) | 16.91 | 17.69 | 19.03 | 17.46 |
| | RH (%) | 51.58 | 49.69 | 46.40 | 50.76 |

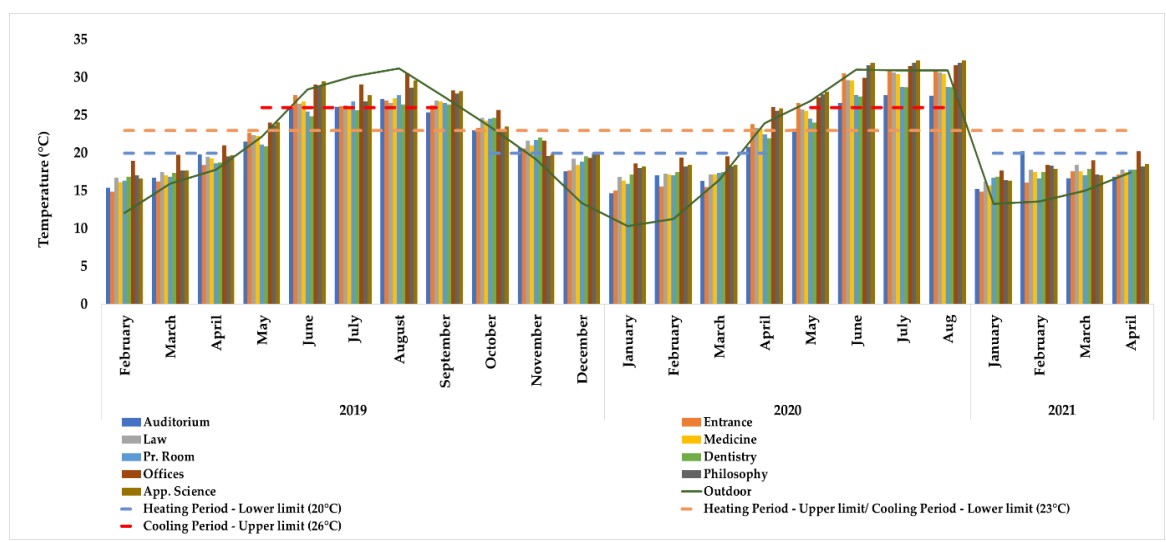

**Figure 4.** Mean monthly average temperature (°C) values within the museum along with the respective limits.

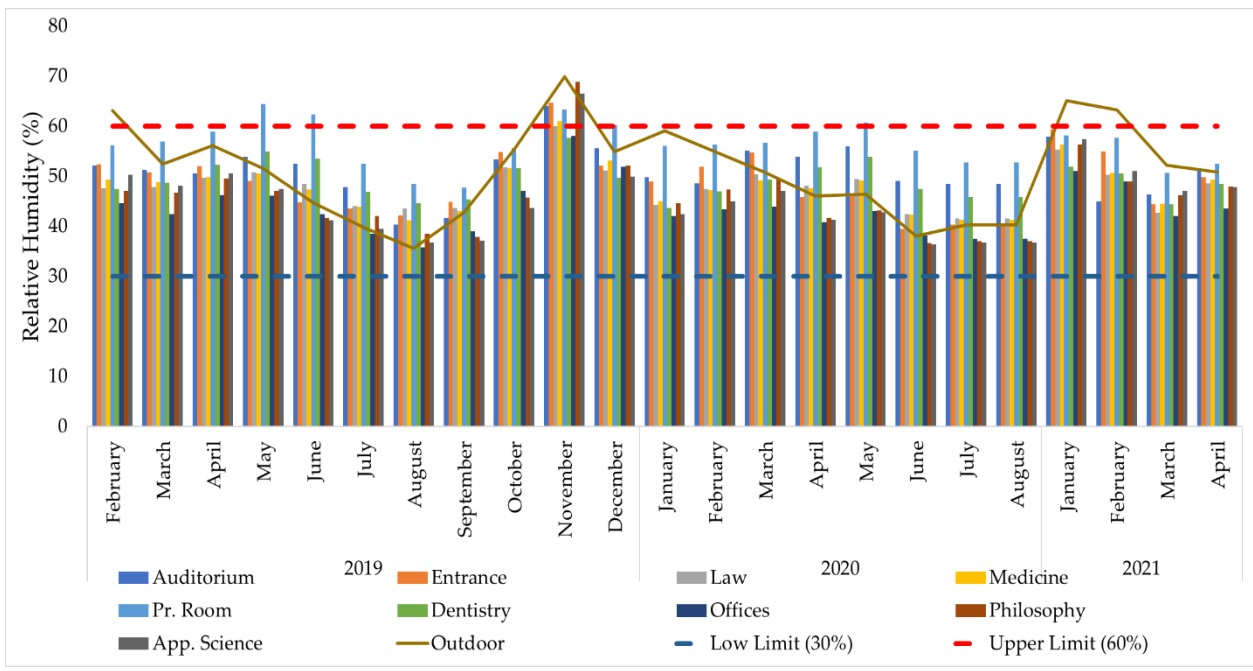

**Figure 5.** Mean monthly relative humidity (%) values within the museum along with the respective limits.

From Figure 4, it is obvious that the offices (2nd floor) present the highest mean monthly T during the cold months, which is mainly due to the frequent use of HVAC split units by the museum's personnel during operating hours. However, during the summer, the air conditioning units do not seem to retain adequate levels of T within the room, as this was later found to be close to the external values of T.

In addition, the average monthly T within the e Applied Sciences and Philosophy exhibition rooms (2nd floor) also demonstrated high values, overpassing the proposed limit range (23–26 °C) during the summer months. As mentioned above, this is mainly because the roof of the second floor absorbs large amounts of solar radiation during the hot days of the Greek summer, resulting in higher internal T values within the lower rooms.

Another interesting result, which applies to all of the rooms under investigation, is that the mean monthly values of T were found to be lower than the proposed limit range (20–23 °C) for the cold months of December, January, February, and March. Furthermore, during the summer months (June, July and August), the indoor T levels approached those

of the respective of the external environment, depicting an insufficient heating–cooling system, poor thermal insulation, and the necessity for deep energy retrofitting in this building.

On the contrary, Figure 5 illustrates that the RH levels within the whole building remained at relatively adequate levels for the tenant's thermal comfort and for the preservation of exhibits, as the majority of measurements did not surpass the respective limit range (30–60%). An exception to this behavior was found during November 2019, when the RH surpassed the limits in many exhibition rooms, such as in the Applied Sciences and Philosophy exhibition rooms (second floor), which was mainly due to frequent local rainfall during that time. In the same month, the mean monthly external RH values were found to be very close to the RH levels inside the building.

Table 2 summarizes the percentages of all of the indoor measurements of T, which were found to be within the proposed limits for the whole building and for each examined room separately. The results are divided into the cold (October–April) and the warm (May–September) monitoring periods of the experiment. It is clear that indoor T does not favor thermal comfort levels, as the percentages for almost all of the cases are relatively decreased. More precisely, more than half of the T values within the comfort range were calculated for the offices of the second floor (51%) during the cold period of 2019. This room generally demonstrates higher percentages compared to the rest of the building, which is mainly because of the frequent use of HVAC systems from the permanent staff during working hours. The inadequate thermal conditions of the building are depicted during the measurements from the cold period, especially for the cases of the Presentation room and the Dentistry room, where the percentage of the T values within the limits was 0%.

**Table 2.** Percentages of indoor temperature measurements within the recommended limit ranges for the entire experimental period.

| Experimental Point | Cold Period (October–April) | | | Warm Period (May–September) | |
|---|---|---|---|---|---|
| | 2019 | 2020 | 2021 | 2019 | 2020 |
| Auditorium | 15% | 5% | 7% | 28% | 19% |
| Entrance | 6% | 4% | 5% | 41% | 15% |
| Law | 18% | 5% | 6% | 45% | 26% |
| Medicine | 15% | 2% | 2% | 37% | 21% |
| Pr. Room | 3% | 0% | 0% | 17% | 28% |
| Dentistry | 1% | 1% | 0% | 45% | 25% |
| Offices | 51% | 40% | 33% | 18% | 25% |
| Philosophy | 20% | 15% | 4% | 37% | 15% |
| App. Science | 20% | 20% | 7% | 27% | 16% |
| Museum (in total) | 17% | 10% | 7% | 33% | 21% |

In total, the museum demonstrated acceptable percentages from 7 to 33% for the entire experimental period, a result that highlights its poor thermal behavior. Furthermore, significant T differences within the museum during the entire year are likely to negatively affect its exhibits. Therefore, renovation actions related to the thermal protection and/or upgrading of the heating–cooling system were considered to be necessary. On the contrary, regarding RH levels, the comfort conditions for the visitors, employees, and exhibits were found to be adequate, as almost all of the measurements were found to be within the respective limit range during the entirety of the experimental procedure. However, it should be noted that RH values are strongly proportional to T fluctuations, meaning that when T decreases, RH tends to increase; the opposite is also true.

### 3.2. Indoor Air Quality

Indoor air quality levels were also investigated in the halls of the museum. $CO_2$ and TVOC concentrations were measured and compared to the ASHRAE Standard 62.1-2010 [38] (American Society of Heating, Refrigerating and Air-Conditioning Engineers) proposed $CO_2$ limit of 1000 ppm and TVOC tolerance limits as reported by Raatikainen et al. [39]. As

depicted in Figure 6, the $CO_2$ levels were found to be within the respective limits, with the Offices (473 ppm) and the Auditorium Hall (472 ppm) on the ground floor depicting the highest average $CO_2$ concentration.

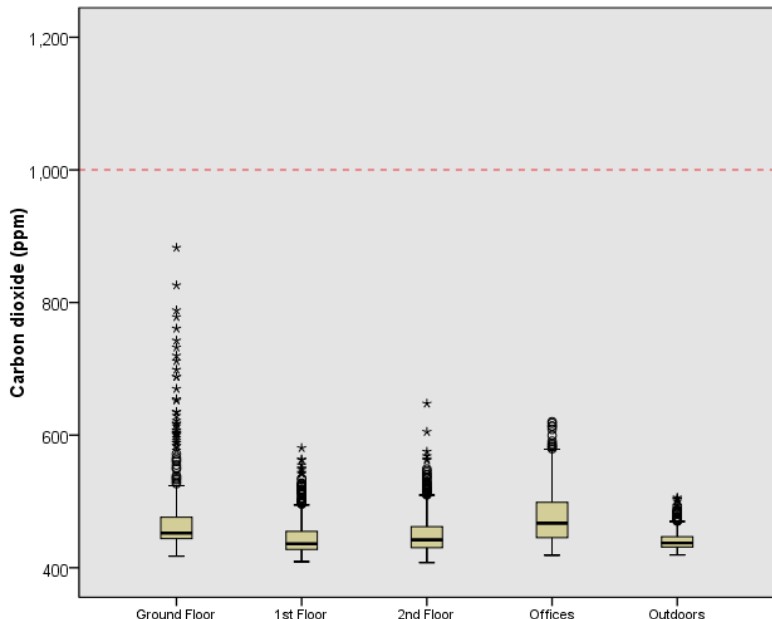

**Figure 6.** Carbon dioxide concentrations in the museum halls and outdoors. ''*'' symbol is used for the outliers.

This observation can be attributed to the fact that the offices were constantly full of the museum's personnel, especially before COVID-19 restriction measures were implemented. What is more, prior the pandemic and during the time between the 1st and the 2nd lockdown, events were often hosted in the Auditorium, and renovations continued to be conducted during the pandemic. Nevertheless, even the outliers that were mostly associated with such events are well below the ASHRAE limits, indicating the lack of a mechanical ventilation system, which would improve the indoor conditions. In the same Figure, one may notice that the outdoor mean concentration of $CO_2$ reached 443 ppm, which is very close to the values of the indoor environment. Thus, natural ventilation (openings) seems to play a key role in the building's microclimatic conditions. Combined with the results of T and RH, a renovation strategy that includes the installation of a state-of—the-art mechanical ventilation system is proposed, not only to retain low levels of $CO_2$ but also to maintain adequate thermal conditions for the occupants by avoiding the frequent opening of doors and windows. The calculation of the air exchange rates with the use of blower door and/or tracer gas methods in each room may be considered as a next step for future monitoring campaigns within this building in order to obtain more detailed results on ventilation rates during different periods.

The average $CO_2$ concentrations in the exhibition halls of both floors were found at 445 ppm and 450 ppm, respectively. Both the Entrance of the museum (450 ppm) and the Applied Sciences hall (450 ppm) recorded the highest concentrations amongst the halls of their respective floors and remained well below the abovementioned limit (Figure 7).

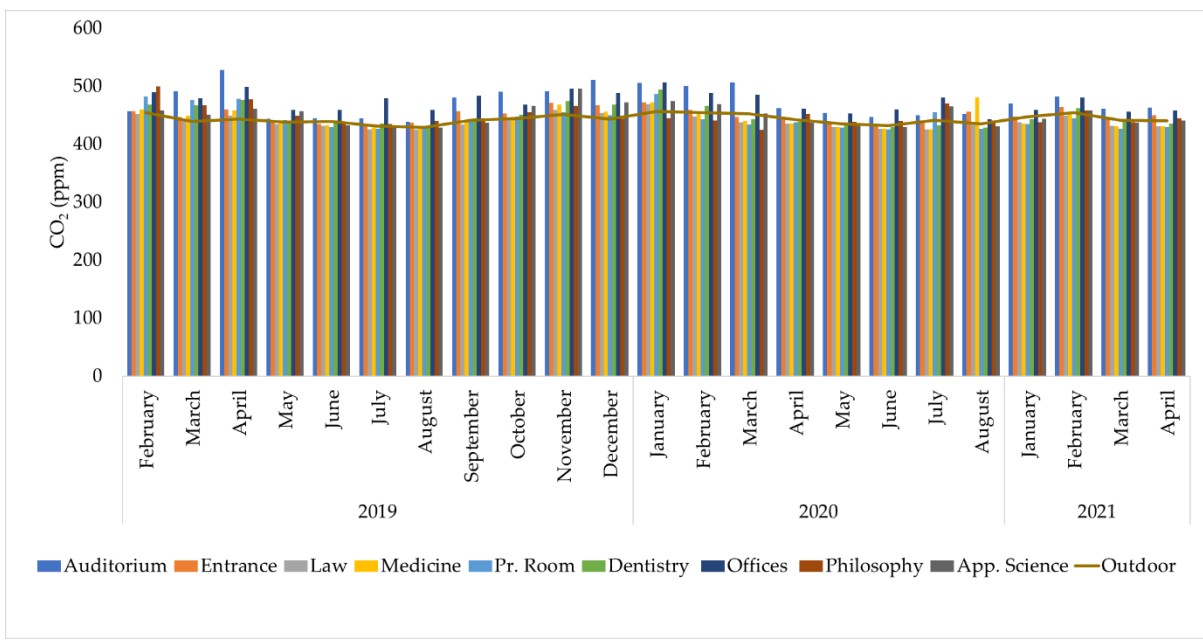

**Figure 7.** Carbon dioxide levels in every hall of the museum for the entire monitoring process.

The first floor recorded the highest concentration of TVOC, with 5 ppm, which can be attributed to the combination of using cleaning products along with the insufficient ventilation of the Presentation Room (11 ppm) and Dentistry Hall (9 ppm). One also may notice that the area with the second highest concentration of TVOC was the Auditorium (5 ppm), which also had the 2nd highest concentration of $CO_2$; however, these levels were still well below the threshold limits (Figure 8).

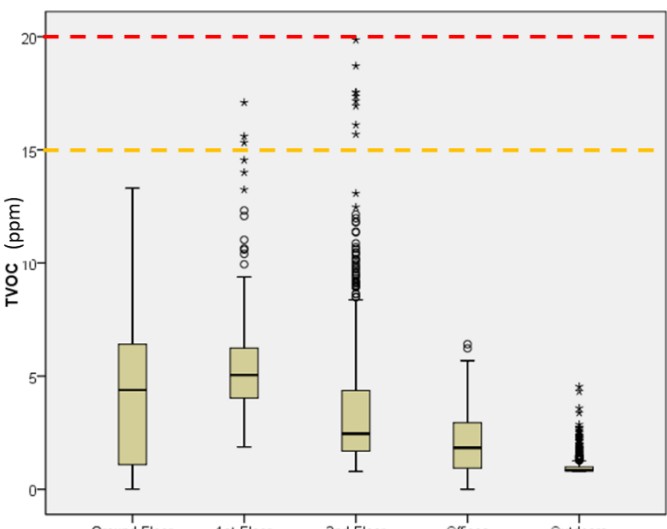

**Figure 8.** Total volatile organic compounds (TVOC) concentration in the museum premises for the entire measurement period. ''*'' symbol is used for the outliers.

However, this was not the case for the 1st and 2nd floors, where there were some cases where the TVOC limits were surpassed. More specifically, as presented in Figure 9, the findings from the Presentation Room show that during June 2020, just after the re-opening of the museum under the strict health protocols due to the COVID-19 pandemic, the TVOC concentration was at its peak (16 ppm). Similarly, the TVOC levels in the Dentistry Hall were considerably high; they did not surpass any of the proposed limits.

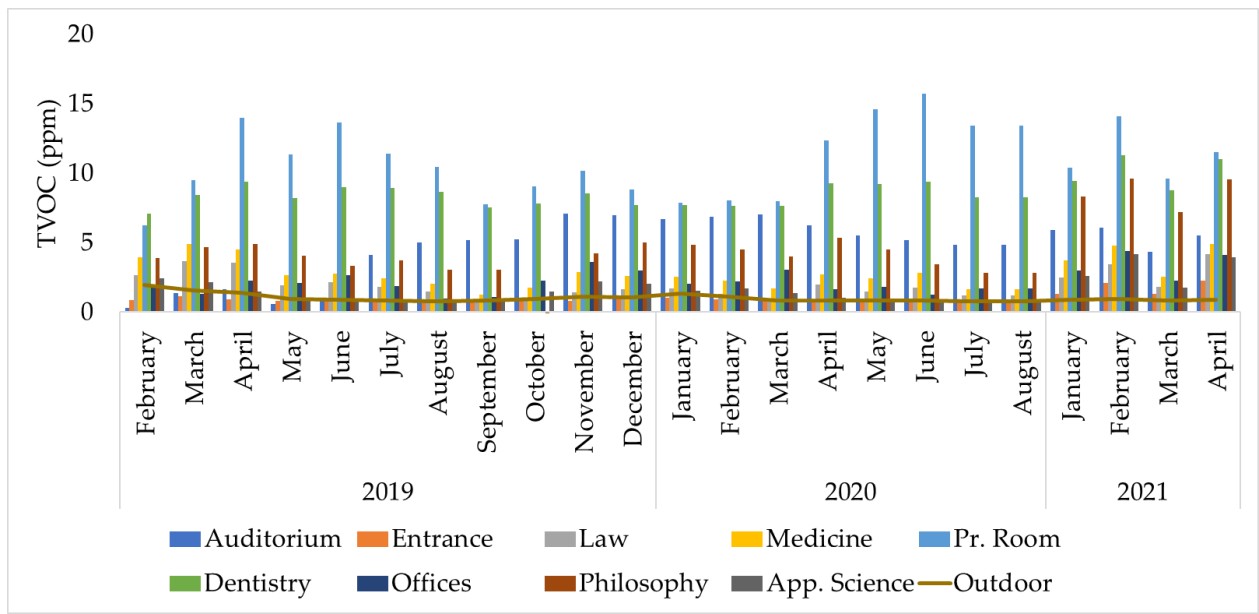

**Figure 9.** TVOC levels in every area of the museum throughout the entire measurement period.

Furthermore, the investigation of the influence of ambient air quality levels to indoor conditions is also of great importance. For this reason, Figures 10 and 11 present the indoor to outdoor ratio (I/O) results for $CO_2$ and TVOC during the entire experimental period at different experimental points. From Figure 10, it is clear that the $CO_2$ I/O ratios are very close to 1 for all of the examined rooms and especially during the warm months, when doors and windows were frequently open. Thus, external $CO_2$ levels seem to have an impact on the internal environment, as the indoor $CO_2$ concentrations were at relatively low levels, as mentioned above. I/O ratios > 1 are related to high occupancy levels within the examined rooms.

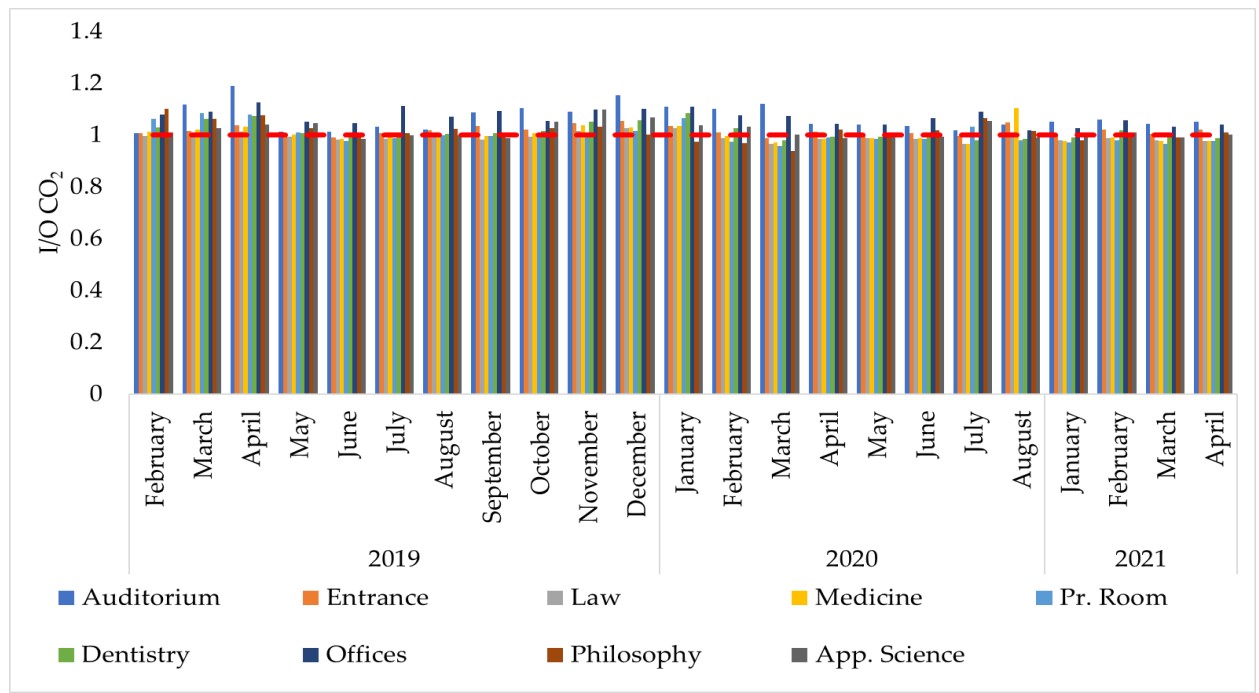

**Figure 10.** Indoor to outdoor (I/O) ratio of $CO_2$ levels within the museum during the entire experimental period.

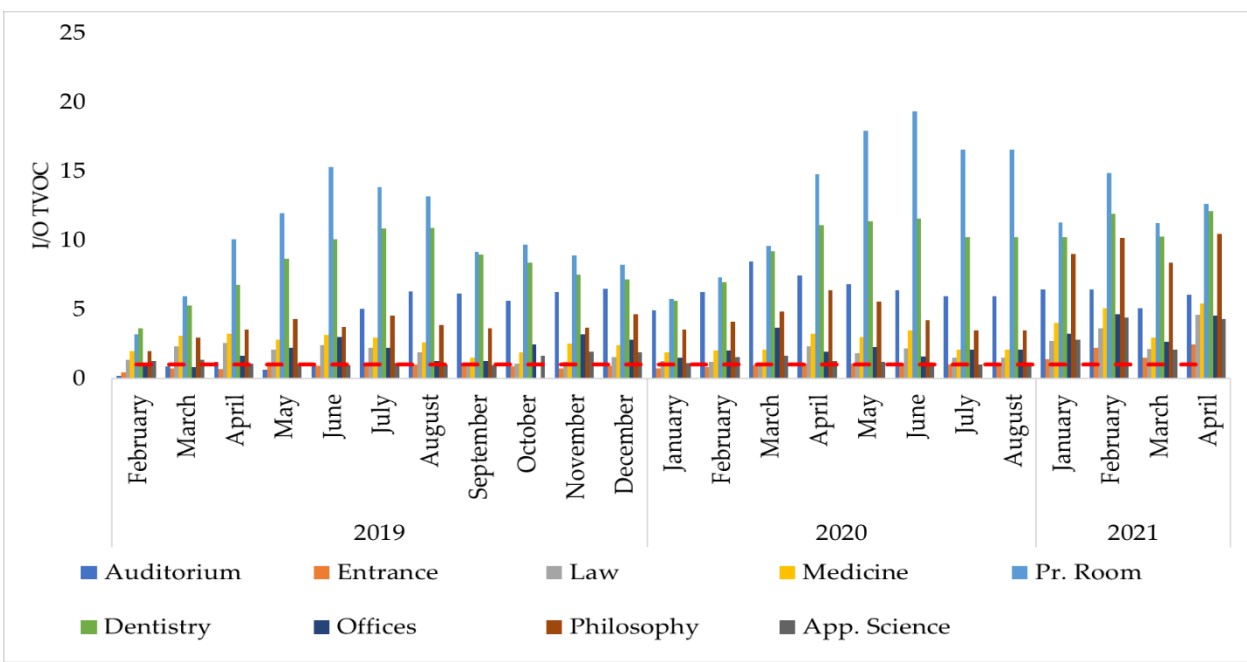

**Figure 11.** Indoor to outdoor (I/O) ratio of TVOC levels within the museum during the entire experimental period.

An opposite behavior is observed for the case study of TVOC. Figure 11 illustrates that during all of the experimental months, the majority of TVOC I/O ratios were found to be >1 in most rooms, even during summer. Ambient levels of TVOC did not demonstrate a serious impact on the microenvironment of the building, as internal emissions of TVOC are frequently related to indoor sources, such as furnishing, wooden materials, coatings, paints, and the daily use of detergents. These TVOC sources were also reported in the research of Mečiarová et al. [40].

*3.3. COVID-19 Restriction Impact*

The impact of the COVID-19 pandemic on the building's behavior is profound in terms of both the $CO_2$ levels as well as the TVOC, as can be seen in Tables 3 and 4. The TVOC concentration, representing the usage of cleaning materials such as detergents and antiseptic hand gels, was higher in both years compared to 2019 levels in almost all areas. The areas that present higher increases are the Auditorium and the exhibition rooms on the 2nd floor, while regarding the whole building, TVOC concentration doubled in 2021 compared to 2019.

**Table 3.** Percentages of differences for TVOC levels within the museum compared to 2019.

| Experimental Point | 2019 vs. 2020 | 2019 vs. 2021 |
|---|---|---|
| Auditorium | 247% | 238% |
| Entrance | 31% | 108% |
| Law | 11% | 57% |
| Medicine | 6% | 42% |
| Pr. Room | 25% | 35% |
| Dentistry | 11% | 27% |
| Offices | −2% | 83% |
| Philosophy | 14% | 145% |
| App. Sciences | 36% | 140% |
| Museum (in total) | 42% | 97% |

**Table 4.** Percentages of differences for $CO_2$ levels within the museum compared to 2019.

| Experimental Point | 2019 vs. 2020 | 2019 vs. 2021 |
|---|---|---|
| Auditorium | −4% | −4% |
| Entrance | −1% | −1% |
| Law | −1% | −2% |
| Medicine | −2% | −4% |
| Pr. Room | −4% | −7% |
| Dentistry | −3% | −6% |
| Offices | −3% | −6% |
| Philosophy | −5% | −4% |
| App. Sciences | −2% | −3% |
| Museum (in total) | −2.8% | −4.1% |

Another indicator illustrating the influence of the COVID-19 restriction measures was the $CO_2$ levels. As $CO_2$ levels are mainly influenced by human presence, it was expected that a decrease in the $CO_2$ levels would be observed compared to the other two years, reaching up to 4.1% for 2021. The Presentation Room recorded the most striking decrease in 2021, which can obviously be attributed to the fact that it was closed from October 2020 until April 2021; the decrease observed in the offices was also important (−6%).

## 4. Conclusions

The particular environmental and climatic conditions that apply in the Mediterranean zone, a region rich in tangible cultural heritage with millions of visitors every year, makes the promotion of the deep energy retrofitting of historic buildings converted to museums a significant challenge, mainly due to vulnerable but delicate nature of the exhibits in those museums. When retrofitting measures are to be applied in such buildings, the pre-intervention evaluation of the indoor environmental quality, a critical factor for the preservation and conservation of exhibits and a determinant for the well-being of the users of the building, is crucial for decision makers regarding the optimal retrofitting strategy to be followed. Therefore, the assessment of the quality of the indoor environmental is a key point in the deep energy retrofitting approaches recommended for the Mediterranean reality. As a result, in this work, the evaluation of the indoor microclimatic conditions in a historic building converted to a museum is presented as part of the greater MedZEB (Mediterranean Zero Energy Building) approach, a concept developed within the HAPPEN Project, in which the concept of nZEB (nearly Zero Energy Building) is adapted to the specific characteristics of the Mediterranean area.

For this reason, an integrated 23-month experimental campaign to evaluate the indoor environmental quality of a museum housed in a historic building located in Athens took place from February 2019 to April 2021, which was divided into two periods and aimed to contribute to the development of an energy retrofitting protocol for museums hosted in historic buildings by providing useful data. The key findings of this campaign can be summarized as follows:

The mean monthly T values were found to be lower than the proposed limit range (20–23 °C) for the cold months of December, January, February, and March in all rooms under investigation, whereas the highest mean monthly T during the cold months were recorded in the 2nd floor offices, which was mainly due to the frequent use of HVAC split units by the museum's personnel. On the other hand, during the summer months, the average monthly T within the Applied Sciences and Philosophy exhibition rooms (2nd floor) demonstrated high values, surpassing the proposed limit range (23–26 °C). As a whole, the museum demonstrated low acceptable T value percentages, which were within limits ranging from 7 to 33% for the entire experimental period, indicating the building's poor thermal behaviour. Moreover, the significant T differences observed inside the museum during the year are likely to have negative effects to the exhibits. Therefore, renovation actions related to the thermal protection and/or the upgrading of the heating–

cooling system, such as the increase of its HVAC power to avoid T levels out of the comfort range, are considered necessary.

With regard to the $CO_2$ levels, they were found to be within their respective limits, with the offices (472 ppm) and the Auditorium Hall (472 ppm) at ground floor recording the highest average $CO_2$ concentrations. In addition, the TVOC concentration, representing the usage of cleaning materials such as detergents and antiseptic hand gels, was higher in 2020 and doubled in 2021 as a result of COVID-19 pandemic protection measures, compared to 2019 levels in almost all areas under investigation. On the contrary, there was a decrease observed in the $CO_2$ levels, with a reduction of 4.1% being reached for 2021. Regarding the indoor air quality of the building, the findings highlight the lack of a mechanical ventilation system, which would, in turn, improve the indoor air quality conditions; thus, the installation of an efficient ventilation system for the building is highly recommended in a future renovation.

Overall, the evaluation of the indoor environmental quality equips decision makers with an extra robust tool to choose the best options among a set of intervention choices in order to achieve a cost-effective retrofitting approach. In this case, study, the installation of a new HVAC system is required in order for adequate levels of thermal comfort to be sustained during the whole year as well as for ventilation requirements to be met for lower levels of gaseous pollutants and fine particulate matter. In addition, any intervention on the internal envelope of the building should promote stable hygrothermal conditions and avoid the emission of volatile organic compounds, which, in turn, could adversely affect the nature of the museum's artifacts as well as the well-being of the building's users.

More specifically, the actions recommended to take place in the museum include: The insulation of the balcony (22 mm thick plywood), the installation of a central heat pump unit (150 kW) with COP = 3 for the specific region, which will cover the cooling, heating, and hot water needs of the museum, and the replacement of the existing deficient lighting system with LED lighting to meet the modern needs of the museum and to save electricity

The aim of this deep retrofitting refers to energy consumption improvements (up to 40%), to ensure adequate thermal comfort and IAQ for the visitors/employees, and to ensure the ideal conditions for the maintenance of exhibits. In order to fully comply with MedZeB targets and to increase energy efficiency and savings, an internal insulation may be needed—stone, if it is properly insulated, helps the efficient regulation of temperature in a home—as well as the implementation of mechanical ventilation.

**Author Contributions:** Conceptualization, C.E. and J.M.S.L.; methodology, N.B.; software, P.T.; validation, N.B., P.T. and C.E.; formal analysis, C.E.; investigation, C.E.; resources, V.N.; data curation, P.T. and V.Z.; writing—original draft preparation, C.E., N.B., J.M.S.L. and T.K.; writing—review and editing, C.E., N.B., V.N. and J.M.S.L.; visualization, C.E.; supervision, M.N.A.; project administration, M.N.A. and J.M.S.L.; funding acquisition, M.N.A. and J.M.S.L. All authors have read and agreed to the published version of the manuscript."

**Funding:** This research has received funding from the European Union's Horizon 2020 research and innovation programme under Grant Agreement No 785072—"Holistic AProach and Platform for the deep renovation of the med residential built Environment" HAPPEN Project.

**Data Availability Statement:** The data presented in this study are available upon request from the corresponding author. The data are not publicly available due to privacy restrictions.

**Acknowledgments:** This project has received funding from the European Union's Horizon 2020 research and innovation programme under grant agreement No 785072. The authors would also like to acknowledge The Athens University Museum's director, board, and personnel for providing permission for this campaign to be held in the building.

**Conflicts of Interest:** The authors declare no conflict of interest.

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
