# Peer review of "Indoor Environmental Quality Evaluation Strategy as an Upgrade (Renovation) Measure in a Historic Building Located in the Mediterranean Zone (Athens, Greece)"

_applsci, doi:10.3390/app112110133_

Round 1
Reviewer 1 Report
The article presented for review is based on the assessment of air quality in the real building (actually used). The assessment of microclimate parameters should definitely be taken into account when deciding on the method and scope of renovation of buildings in terms of energy. However, it must not be forgotten that the performance of the existing HVAC systems operating in the building, their current energy requirements, etc. must also be taken into account when assessing such an assessment.
The potential of the article is large. However, the article needs some minor changes.
The work concerns a case study for indoor air quality, but absolutely does not refer to renovation possibilities, especially in terms of HVAC installations.The article does not actually discuss the existing HVAC installations, it only assesses the parameters of the microclimate.The title of the article should be changed.
The article states that the new HVAC systems should provide appropriate microclimate conditions for both people and exhibits. However, it was not specified what these conditions are especially in relation to the exhibits, which should be treated as a priority here.
Row 138: There is „Moreover, the building in naturally ventilated with open doors…” It shuld be: Moreover, the building is naturally ventilated with open doors …”
Row 215: Average air temperature values are given. What were the extreme values measured (min. and max.)
Row 218: There is „hot summer months the rooms of the third floor”. I think is second floor.
Rows 288-295: It should be remembered that very often RH is closely related to a specific range of temperature values and we should not only consider it separately, but above all together.
Row 303: There seems to be an erroneous reference to literature here. Instead of [36] it should be [37]. Was it possible to change the scope of the TVOC on the basis of one document?
Row 304: The CO2 level (472 ppm) is rather low. It is difficult to assess this value here if the value of the background level, i.e. the outside air, is not known. Such low values may indicate long-term open windows and doors, which is also confirmed by the T and RH results. This means that natural ventilation (airing) has a decisive influence on the microclimate conditions. How does this relate to the need to renovate the layout and the building?
Fig. 9: Since the other charts listed a range of max. and min. values, why wasn't this done in this chart? This should be completed.
Conclusion should be corrected. In the current version, it is a repetition of the discussion of the results and does not actually refer to the possibility of renovating a building or installation. The conducted research shows a lot of information that is not provided here.
In the article there is lack of "." at the end of many of the sentences. For example in lines 61 and 64 (after [18] and [20]).
Reviewer 2 Report
The work is original, clear, appealing and well-written; the conclusions are supported by the results and I think that the topic discussed is very actual and interesting for readers from different technical areas. However, I have some questions and I suggest to publish it only after some revisions.
1) The authors discuss 320-days of campaign but the months discussed are more (22 months)...something is wrong or not clear. Please, clarify this point.
2) Building description: the stone/s used for the building of the museum could affect temperature, umidity, ventilation and so on of the museum itself due to its natural properties such as porosity, ability to humidity absorption, ability to release water vapour, dimensional properties and so on..but this data is missing and therefore not discussed. How the building itself contributes to dissipate or store heat, umidity and air?
3) line 138, the auhtors said: the building is naturally ventiled with open doors and windows". The authors should discuss the influence of outdoor air quality onto indoor air quality because the pollution of the area around the building of course influences the air quality inside.
Round 2
Reviewer 1 Report
Thank you for considering my comments. The changes introduced in the text significantly influenced the quality of the article.
Reviewer 2 Report
All my remarks were revised and the quality of the paper is high.